# Modulation of Fat Deposition–Gut Interactions in Obese Mice by Administrating with Nobiletin

**DOI:** 10.3390/genes14051062

**Published:** 2023-05-10

**Authors:** Cunzhen Zhao, Jiahua Guo, Chunyu Du, Yongjie Xu

**Affiliations:** 1College of Life Science, Xinyang Normal University, Xinyang 464000, China; 2Institute for Conservation and Utilization of Agro-Bioresources in Dabie Mountain, Xinyang Normal University, Xinyang 464000, China

**Keywords:** nobiletin, fat deposition, intestinal microflora, metabolic pathways, correlation analysis

## Abstract

Intestinal microflora is correlated with obesity, metabolic diseases and digestive tract dysfunctions that are closely related to human health. Nobiletin (NOB) is a dietary polymethoxylated flavonoid with protective effects and activities against oxidative stress, inflammation and cardiovascular disorders. However, the effect and molecular mechanism of NOB in regulating white fat deposition have not been explored. In this study, we reported that NOB administration attenuates weight gain and glucose tolerance in mice fed a high−fat diet (HFD). Additionally, NOB administration substantially restored lipid metabolic disorder and repressed the level of genes related to lipid metabolism in HFD−induced obese mice. The sequencing of 16S rRNA genes in fecal samples unveiled that NOB administration reversed HFD−induced intestinal microbiota composition, particularly in the relative abundances of Bacteroidetes and Firmicutes at the phylum and genus level. Furthermore, NOB supplementation significantly improved the indexes of Chao1 and Simpson and implied NOB can improve intestinal flora diversity in HFD−fed mice. Next, we used LEfSe analysis to explore biomarkers presented as a taxon in different groups. Compared to the HFD group, NOB treatment significantly diminished the proportion of *Ruminococcaceae, Ruminiclostridium, Intesinimonas, Oscillibacter* and *Desulfovibrio*. Enriched metabolic pathways were predicted by Tax4Fun analysis and demonstrated that the lipid metabolic pathway is higher in the HFD + NOB group. More importantly, the correlation analysis demonstrated that *Parabacteroides* was significantly positive and *Lactobacillus* was negatively related to both body weight and inguinal adipose tissue weight. Collectively, our data emphasized that NOB has the potential to attenuate obesity and confirmed a mechanism for gut microbiota that mediated the beneficial effect of NOB.

## 1. Introduction

Obesity makes people more susceptible to numerous diseases including diabetes, cardiovascular diseases, cancer and some immune−related disorders. The global obesity epidemic and the awareness of the menace of obesity on population health can be noticed in the excessive national reports and policies from numerous countries. The growing contribution of obesity leading to significant economic costs as well as public health challenges and its burden of malnutrition is a threat to national development [1]. Obesity mainly ascends from an imbalance between energy intake and expenditure with excessive fat deposition in the body [2]. Lipid metabolism confers an essential role to adipose tissue in organism homeostasis and lipid metabolic dysfunctions will lead to obesity and the related type 2 diabetes mellitus (T2DM)) or WAT paucity [3].

The relationship between human health and gut microbiota has attracted much attention in the last decades. Intestinal microbiota dysbiosis has been associated with obesity, metabolic diseases, liver pathologies and digestive tract dysfunctions [4,5]. Enormous evidence identified that the gut microbiota can affect the functions of the central nervous system (CNS) as well as host behavior, and other unknown functions are also being explored [5]. Research shows that the influence of gut microbiota on obesity is mainly attributed to two aspects. On the one side, the gut microbiota has the capacity to degrade indigestible polysaccharides into absorbable monosaccharides and lead to an increase in hepatic lipogenesis in the host. On the other side, gut microbiota restrains the intestinal gene level of *ANGPTL4* to regulate LPL activity and further increase cellular fatty acids uptake and the accumulation of adipocyte triglycerides [6,7]. Moreover, obese mice and humans had a higher Firmicutes/Bacteroidetes ratio, implying that obesity converts the nature of the gut microbiota [8,9]. Furthermore, recent studies also found that alterations in microbiota composition at lower taxonomic levels have been correlated with obesity and the abundance of genera or even specific bacterial species may better clarify the dysbiosis correlated with obesity [10,11,12].

Nobiletin (NOB; 3′,4′,5,6,7,8−hexamethoxyflavone) is a dietetic polymethoxylated flavonoid dissociated from citrus fruits with the protective effects activities against oxidative stress, inflammation, cancer progression, metabolic and cardiovascular disorders, and aging and age−related neurodegeneration [13,14,15]. Studies have shown that nobiletin exerts anti-adipogenic effects through modification of the AMPK signaling pathway and prompts brown adipocyte-like phenotype in 3T3−L1 cells [16,17]. Although nobiletin also has been demonstrated to stimulate lipolysis via activating signal cascades and which is mediated by cAMP/CREB [18], the effect and mechanism of nobiletin in regulating white fat deposition have not been explored. In this study, we explored the function of nobiletin in regulating white fat deposition and blood glucose and lipid levels in HFD−fed mice. Additionally, the impact of nobiletin on the gut microbiota has been examined by analyzing the changes in microbial composition and abundance. These findings provide an anti−obesity mechanism of nobiletin and scientific evidence to exploit nobiletin-based functional foods.

## 2. Materials and Methods

### 2.1. Animals

A total of 32 male Kunming mice weighing 18–20 g mice and aged 6 weeks were purchased from Wuhan Shubeili Bioscience Co. Ltd. (Wuhan, China). Mice were housed in a room temperature and controlled environment to gain food and water in freedom. Normal diet (chow, 10% of calories from fat, 70% and 20% of calories were, respectively, from carbohydrate or protein; XTCON10−1) and a high-fat diet (HFD, 60% of calories from fat, 18% of calories from protein and 22% of calories from carbohydrate, XTHF60−1) were purchased from Jiangsu Xietong Pharmaceutical Bioscience Co. Ltd. (Nanjing, China). Nobiletin (NOB, CAS No. 478-01−3) was purchased from Sichuan Weikeqi Biological Technology CO., LTD (Chengdu, China). All animal experimental procedures were approved by the Ethics Committee of Xinyang Normal University (protocol code XYEC−2021−011 and 1 January 2021 of approval).

### 2.2. Preparation and Grouping of Animal Models

All Kunming mice aged 6 weeks were acclimatized to the environment for 2 weeks prior to the start of the test, then divided into two groups separately fed a normal diet (chow) or a high−fat diet (HFD) randomly until the weight of the HFD group was significantly higher than the chow diet group for 7 weeks. On this basis, the mice were further divided into four groups: control group (chow + Veh), nobiletin group (chow + NOB), high-fat control group (HFD + Veh) and high-fat nobiletin group (HFD + NOB). NOB was dissolved by 0.5% (sodium carboxymethyl cellulose, CMC−Na) and mice were treated by intragastric gavage at a dose of 100 mg/kg body weight and at the same time every other day. Control mice were administrated by an equal volume of vehicle (0.5% sodium carboxymethyl cellulose, CMC−Na). Mice were administrated by gavage for 6 weeks and body weight and food intake were valued weekly. Fecal samples were gathered from individual mice after 6 weeks of treatment and quickly stored at −80 °C for subsequent fecal DNA extraction and 16S amplicon sequence analysis. Organs and tissues were collected and weighed for further HE staining, as well as frozen at −80 °C for RNA extraction.

### 2.3. Hematoxylin and Eosin Staining

Mouse liver, inguinal adipose tissue (IAT) and subcutaneous adipose tissue (SAT) were fixed with 10% paraformaldehyde solution at room temperature for 24 h, sliced and embedded in paraffin. Tissues were sectioned at 4 mm thickness and stained with eosin hematoxylin (H&E). All the stained sections were observed by microscopy.

### 2.4. Glucose Tolerance Tests and Serum Levels of TC and TG Analysis

The mice were fasted for 16 h for a glucose tolerance test (GTT) and a basal blood sample was collected from the tail vein (0 min) after treatment with NOB for 6 weeks. The blood glucose level was measured using a glucometer (Yuwell 590, Nanjing, China). The mice were then administered glucose (2 g/kg body weight) by intraperitoneal injection. Tail vein blood was measured at 30, 60, 90 and 120 min after injection. For serum levels of total cholesterol (TC) and total glyceride (TG) tests, mice were anesthetized with diethyl ether for heart blood collection and serum was obtained by centrifugation. Blood lipids, cholesterol and other indicators were measured by a commercial kit (Nanjing Jiancheng Bioengineering Institute, Nanjing, China).

### 2.5. RNA Extraction and RT-qPCR

Total RNA was extracted from white adipose tissue or epithelial cells using TRIzol (Takara, Japan) and converted to cDNA with reverse transcriptase (Takara, Japan) on the basis of the manufacturer’s protocol. Real−time PCR (qPCR) analysis was executed using SYBR Green PCR Master Mix (Takara, Japan) and analyzed with a Real−time PCR System (Bio−Rad, CFX96). Relative expression of mRNAs was determined after normalization to GAPDH. All real−time qPCR reactions were carried out in triplicate. Primer sequences are listed in Table 1.

### 2.6. 16S rRNA Gene Sequence Analysis

The genomic DNA of mice feces was isolated by the SDS method. The purity of extracted DNA was tested by agarose gel electrophoresis and diluted to 1 ng/μL with sterile water. V3–V4 hypervariable region of 16S rRNA genes to amplify the DNA sequences.

Library construction was adopted by Truseq ^®^ DNA PCR free sample preparation kit. Qubit and Q−PCR were used to quantify the constructed library and the NovaSeq 6000 system was used for sequencing.

The reads of each sample were spliced with FLASH after the barcode and primer sequences are truncated [19]. Next, contrastive analysis between tags sequences and the species annotation database were explored. Then, removing chimeric sequences and the final effective tags were obtained [20,21]. The operational taxonomic units (OTUs) were clustered and annotated at a 97% similarity threshold. Following that, the rarefaction curve analysis was displayed using the R software (Version 2.15.3). The analysis of α−diversity and β−diversity containing the Chao1, ACE, Simpson and Shannon indexes, as well as PCA and PCoA, was performed by QIIME software (Version 1.9.1). Additionally, R software was used to determine the differences in α−diversity and β−diversity index between groups. The linear discriminant analysis (LDA) effect size (LEfSe) was used for selecting biomarkers among different groups and the filter value of LDA Score at >2.

### 2.7. Statistics Analysis

All data were exhibited as the mean ± SEM. Significance test between groups was performed by using Student’s t−test or ANOVA (for comparison of three or more experimental conditions) and figures were performed by GraphPad Prism 6 software. Significance was determined at *p* < 0.05 or *p* < 0.01.

## 3. Results

### 3.1. NOB Supervision Depresses HFD-Induced Obesity

To investigate the physiological function of NOB in adipose tissue and energy homeostasis, we generated an obese mouse model using high−fat diet feeding for 6 weeks. The weight of mice in the high−fat group was increased by 14.11% and was significantly higher compared to the chow−fed group (Figure 1A,B). Then, mice were treated with a dose of 100 mg/kg of NOB or vehicle by garage every other day and sustained for 6 weeks. With the extension of treatment time, NOB treatment attenuated the body weight gain of HFD−fed mice and gained less body weight than control mice under HFD conditions or chow-fed conditions until the sixth week (Figure 1B–D).

### 3.2. NOB Ameliorates Hyperlipidemia and Lipid Metabolic Disorder in HFD-Induced Obese Mice

Next, we evaluated the effects of NOB on GTT and serum lipid profiles. The results demonstrated that mice fed HFD showed glucose intolerance and the blood glucose declined markedly faster in NOB−treated HFD mice. However, NOB did not show any discernible effects on mice under chow feeding (Figure 2A). The levels of serum TG, TC and LDL−C were increased in HFD-fed mice in contrast to chow-fed mice (Figure 2B). In addition, levels of TG, TC, HDL−C and LDL−C in NOB-treated HFD−fed mice were significantly decreased compared to the HFD-fed mice (Figure 2B), whereas these parameters have no obvious change between NOB−treated chow mice and chow−fed mice, except for the decrease in TG level (Figure 2B).

Furthermore, the subcutaneous adipose tissue (SAT) and inguinal adipose tissue (IAT) weighed less and were smaller in NOB−treated mice than in mice fed a HFD or that are chow−fed, respectively (Figure 2C,D). In agreement with these results, histological analysis showed that NOB−treated mice had markedly reduced lipid contents and adipocyte size in SAT and IAT (Figure 2E,G). These results indicated that NOB administration can inhibit adipocyte hypertrophy of WAT. Even then, we assessed the mRNA expression of lipid adipogenesis genes and lipogenesis genes. The results demonstrated that NOB administration significantly depressed the levels of *PPAR*−*r*, *aP2*, *C/EBPa*, *FASN* and *SREBP*−*1C* in both SAT and IAT in HFD mice, whereas there was no significant change in the chow−fed group (Figure 2F,H). Together, these results illustrate that NOB administration protects mice from HFD−induced hyperlipidemia and fat deposition.

### 3.3. NOB Administration Regulates the Composition of Intestinal Flora in Mice

To explore whether NOB administration induces changes in the community structure of gut microbiota, we isolated genomic DNA from fecal samples of mice and profiled them with the sequencing of the V3–V4 region of the 16S rDNA gene. An average of 89,079 raw reads was generated from each sample. An amount of 1,732,758 clean tags were subjected to OTUs clustering and species classification analysis after removing the low−quality and short sequence length sequences (Appendix A). The Venn diagram for the distribution of all operational taxonomic units showed that 515 OTUs existed in all mice, and 659, 599, 641 and 611 OTUs were found in the group of chow, chow + NOB, and HFD and HFD + NOB mice, respectively (Figure 3A). The ratio of Firmictutes to Bacteroidetes was higher in obese individuals and the increased abundance of Bacteroidetes related to the percentage loss of weight [22]. We observed at the phylum level that the ratio of Firmicutes to Bacteroidetes was definitely raised by feeding a HFD compared to the chow group, whereas NOB supervision ameliorated the Firmicutes/Bacteroidetes ratio compared with the HFD group and lowered levels of Proteobacteria (Figure 3B). Similar observations were also shown at the genus level for *Lactobacillus* and *Bacteroides* (Figure 3C).

### 3.4. NOB Administration Improves the Intestinal Flora Diversity of Mice

Rarefaction curve analysis showed that the curve tended to be flat as the number of sequences increased and covered the rare new phylotype and the most bacterial diversity (Figure 4A). Rank abundance results showed that the curve of the HFD group was shorter and steeper than the chow group and NOB−treated HFD group in the horizontal direction (Figure 4B), suggesting that the species richness in the HFD group declined and that the species uniformity was low, whereas NOB treatment could partially reverse this tendency. Meantime, HFD−fed mice displayed a lower abundance of microbiota, as identified by the reduced Chao1 index compared to the control group (*p* < 0.01). Interestingly, HFD−induced reduction in the Chao1 index was reversed by NOB administration (Figure 4C). Similarly, the trend of the ACE index was consistent with Chao1, but the increment did not achieve a statistical difference (Figure 4D). The Simpson index was generally used to reflect microbial diversity. In our results, the Simpson index of the HFD group was remarkably lower than that of the NOB−treated HFD group (Figure 4E). Meanwhile, the NOB administration tended to restore the reduction of the Simpson index induced by HFD. This implies that NOB administration improves the α−diversity of intestinal flora. To further examine the degree of singularity in β−diversity, the intestinal microbiota between groups was measured by principal coordinates analysis (PCoA). As shown in Figure 4F, the chow and HFD−fed mice displayed a different clustering of microbiota composition, and the NOB under HFD feeding had a distinct structure to their corresponding chow group. In general, the NOB under the HFD feeding group showed a high abundance and diversity to that of the control group.

### 3.5. NOB Administration Attenuates HFD−Induced Gut Microbial and Modified Metabolism Pathway

Furthermore, we performed (LEfSe) analysis coupled with linear discriminant analysis (LDA) to explore specific biomarkers of gut microbiota achieved with the nonparametric factors Wilcoxon test. The results presented that there were significant differences in abundance between NOB−treated and control mice. As shown in Figure 5A,B, 36 populations were demonstrated as the key variables for separating gut microbiota under different treatment groups (the logarithmic LDA score > 2.0), in which 18 of the OTUs (2 were assigned to Peptococcaceae, 7 to Ruminococcaceae and 6 to Proteobacteria) were shown in NOB-treated HFD mice group (Figure 5B). Next, 8 OTUs were displayed in the HFD group (3 were assigned to Peptostreptococcaceae, 2 were assigned to Veillonellaceae and 3 were assigned to Xanthobacteraceae) (Figure 5B), 8 OTUs were displayed in the chow diet group (2 were assigned to Muribaculaceae, 3 were assigned to Bacteroidales and 2 were assigned to Bacteroidia). Two OTUs were displayed in the NOB−treated chow group (1 belonging to phylum Bacteroidetes and 1 belonging to phylum Proteobacteria). Collectively, the intestinal flora of the chow−fed mice mainly belongs to phylum Bacteroidetes and the NOB−treated HFD mice, as well as HFD mice, belong to phylum Proteobacteria, whereas the NOB-treated chow mice have no distinct changes. These results showed that NOB administration could lead to various changes in the gut microbial composition of mice.

In order to investigate the intestinal flora of significant difference between groups, a T−test was used to testify the contrast at otherness classification levels (*p* < 0.05). In contrast to the HFD group, NOB treatment significantly reduced the proportion of *Ruminococcaceae, Ruminiclostridium*, *Intesinimonas*, *Oscillibacter* and *Desulfovibrio* in the genus level (Figure 6A). To further explore the potential function of the intestinal flora within HFD and HFD + NOB groups, enriched metabolism pathways from the KEGG database per group were predicted by Tax4Fun analysis (*p* < 0.05, Figure 6A). In particular, the abundance of lipid metabolic pathways is higher in the HFD + NOB group. Other metabolic pathways, containing carbohydrate metabolism, amino acid metabolism and signal molecules interactions were enriched in the HFD + NOB group. Additionally, signal transduction and genetic information processing were raised in the HFD group (Figure 6B). These results suggest that NOB administration may have generally altered the gut microbiota at the genus level and metabolic pathways.

### 3.6. NOB Administration Converts the Correlation between Fat Deposition and Intestinal Flora

At last, we clarified the linear correlation between adipose tissue and intestinal flora as well as whether NOB treatment changes the correlation. As Figure 7A,B show, in the genus level of the HFD group, *Parabacteroides* was significantly positively correlated with inguinal adipose tissue weight and total body weight using the Spearman analysis (*p* < 0.05), whereas *lactobacillus* displayed an opposite tendency both in body weight and IAT (Figure 7A,B). In addition, *Alloprevotella* was positively related to body weight in the chow group and *Elusimicrobium* was positively related to body weight in the chow and HFD-NOB group (*p* < 0.05, Figure 7A). Notably, *Muribaculum* was extremely correlated with body weight in the HFD−NOB group and in the HFD group, which displayed the opposite trend with body weight, although the relationship is not statistically significant (*p* < 0.01, Figure 7A). Furthermore, *Roseburia*, *Clostridiales*, *Odoribacter* and *Romboutsia* were a significantly negative correlation with IAT in the chow group (*p* < 0.05, Figure 7B). More importantly, the relevance of *Enterococcus* and *Helicobacter* was extremely higher in the HFD−NOB group than those in the HFD group (*p* < 0.01, Figure 7B). Collectively, these results amplified that *Parabacteroides* was significantly positively related and *lactobacillus* was negatively related to both body weight and inguinal adipose tissue, as well as NOB treatment which significantly changes the intestinal flora that affects inguinal adipose tissue and lipid deposition.

## 4. Discussion

The present study provides insights into the role of nobiletin in restraining obesity in obese mice and ameliorating the composition of gut microbiota. The current study demonstrated that the oral governance of NOB at a dose of 100 mg/kg attenuates weight gain, hyperlipidemia and lipid metabolic disorder as well as key adipogenesis−specific markers in HFD−fed mice. In addition, we showed that the beneficial effect of NOB depends upon gut microbiota, mainly presented as follows. Firstly, NOB regulated the composition of intestinal flora in obese mice and changed the gut microbiota structure as well as reduced the intestinal ratio of Firmicutes to Bacteroidetes. Furthermore, NOB improves the intestinal flora diversity and abundance of obese mice. Moreover, NOB administration modified the metabolism pathway and the correlation between fat deposition and intestinal flora.

Extensive evidence indicates that the prevention of adipogenesis is one of the targets for lipid mass reduction in obese individuals. Flavonoids have been found to show potential benefits in the inhibition of obesity and can be observed in abundant quantities in dietary fruits, vegetables, tea and wine [23]. For instance, quercetin is an effective anti−obesity biomolecule that is mainly manifested by inhibiting intestinal starch digestibility and hepatic glucose production as well as protecting against pancreatic islet damage [24,25,26]. The anti−obese properties of kaempferol were influenced by *SREBP−1C* and *PPAR−γ* modulation through AMPK stimulation [27]. Similarly to kaempferol, for intervention and control of obesity and diabetes, resveratrol was identified to up−regulate fatty acid oxidation by regulating the AMPK phosphorylation level. In addition, kaempferol can also increase glucose uptake by GLUT4 translocation and through the up-regulation of SIRT1 to suppress lipid accumulation [28,29]. NOB is one of the flavonoids and, in our study, we demonstrated that NOB decreases the weight of the subcutaneous adipose tissue (SAT) and inguinal adipose tissue (IAT) and ameliorates blood glucose and lipid levels in obese mice. Moreover, NOB reduced the fat deposition that manifested by down−regulating the expression of adipogenesis makers such as *PPAR−r*, *aP2*, *C/EBPa*, *FASN* and *SREBP−1C* in both SAT and IAT in the NOB group. Consistent with our results, previous studies have shown that NOB markedly prevented lipid accumulation and blocked the expression of adipogenic transcription factors in 3T3−L1 cells [15]. Collectively, the above results demonstrated that NOB treatment could restrain HFD−induced obesity.

Another novel finding from our research is that NOB can effectively ameliorate intestinal flora disturbance in HFD−induced mice. Obesity alters the composition of gut microbiota mainly characterized by a high ratio of Firmicutes/Bacteroidetes [8,22\]. In agreement with this, we revealed that NOB administration is able to increase the abundance of Bacteroidetes and reduce the Firmicutes/Bacteroidetes ratio in the present study. Meanwhile, our study also demonstrated an enhanced ratio between *Lactobacillus* and *Bacteroides* at the genus level. These findings alongside the changes in microbiota composition at lower taxonomic levels may better clarify the correlation with obesity [9,10,11]. Through 16S ribosomal RNA gene sequencing, community richness was measured by the Chao1 and ACE indexes, as well as community diversity, which was determined by Shannon and Simpson [30]. What is more, we identified that obese mice displayed a reduced index of Chao1 and Simpson. Additionally, NOB−treated HFD mice tended to restore the tendency of the Chao1 and Simpson index. Although the results of ACE and Shannon have not shown a significant tendency, we still proved that NOB had a marked influence on α- diversity in HFD−fed mice.

Species difference significance analysis found that NOB administration decreased the proportion of *Ruminiclostridium*, *Intesinimonas*, *Oscillibacter* and *Desulfovibrio* in obese mice. These genera have been demonstrated to be positively correlated with obesity and other related disorders. For instance, some herbal extracts such as resveratrol, polyphenol−rich oolong tea and phlorizin treatment could decrease the abundance of *Desulfovibrio*, *Ruminiclostridium* as well as *Intesinimonas* in HFD-induced mice [31,32,33]. *Desulfovibrio* is a sulfate−reducing bacterium that can produce LPS and induce related inflammation [34]. LPS−activated TLR4 increases intestinal permeability and induces a chronic subclinical inflammatory process, leading to insulin resistance and obesity [35]. In our study, we found a decreased *Desulfovibrio* level in NOB−treated HFD mice, which may imply that NOB can decrease LPS levels and play a role in improving intestinal permeability. Consistent with our results, studies have identified that an increased abundance of *Ruminiclostridium* was discovered in the fecal microbiota of obese mice in contrast to the lean counterparts [31,32]. Intriguingly, the function and molecular mechanism of *Oscillibacter* and *Ruminiclostridium* in obesity were still unclear. Therefore, the specific effects of *Oscillibacter* and *Ruminiclostridium* on obesity need to be further investigated.

Consistent with the alteration of gut microbial structure, we detected an improvement in intestinal microflora gene function after NOB treatment. Intriguingly, an increased abundance of carbohydrates, amino acids and lipid metabolism was observed in the fecal microbiota of NOB mice compared to the obese counterparts. Carbohydrates are fermented by microbial to produce short−chain fatty acids (SCFA) such as butyrate, propionate and acetate, which can be utilized by the host [36]. The importance of other SCFAs that are generated by resident species in the intestinal flora including 2−methyl butyrate, isovalerate, valerate and formate cannot be ignored. Our results show that NOB administration improves gut macrobiotic diversity, which may alter some types of SCFA to enhance carbohydrate metabolism and further accelerate obese fat degradation, while the specific types of SCFA caused by bacteria need to be further investigated. The altered distribution of free amino acids in the gut microbiota indicates that the resident species of the gut microbiota are crucial for the amino acid homeostasis of the host [37]. In particular, bacteria of the *Clostridium* genus, which are the fundamental bacteria for lysine or proline utilization and have been identified as key drivers for amino acid fermentation, while bacteria belonging to the *Peptostreptococcus* genus are the main driver for glutamate or tryptophan utilization and the genera *Fusobacterium*, *Bacteroides* could play a prominent role in amino acid metabolism [38]. Consistent with the LEfSe results, we verified that NOB treatment significantly up−regulated the bacteria of the *Clostridium leptum* and the bacterium species in obese mice and which may drive the glutamate or tryptophan use to contribute to the amino acid metabolism. Lipid metabolism is mainly modified by nutrients such as fatty acids and sugars and recent reports have revealed that lipid levels are accompanied by the gut microbiota composition [39]. Studies revealed that *Lactobacillus curvatus* alone or in conjunction with *Lactobacillus plantarum* alleviated cholesterol in plasma and liver and *Lactobacillus* strains had a synergistic effect on hepatic lipid metabolism [40]. Similarly, *Bifidobacterium* spp. diminished levels of circulating plasma triglycerides and LDL and increased levels of HDL in HFD−diet mice [41]. Collectively, we verified NOB treatment can modulate the gut microbiota constitution and reinforce pathways involved in metabolite generations in HFD-fed mice.

The innovative finding from this study is that the gut microbiota genera were directly correlated with body weight and inguinal adipose tissue weight. In our study, the *Parabacteroides* was positively correlated with body weight and inguinal adipose tissue weight on genus level in HFD−fed mice, whereas the correction level is reduced in the HFD−NOB group and reached to a level consistent with the chow mice. Previous reports identifying that the abundance of *Parabacteroides* was markedly increased after a low-carbohydrate diet and a correlational analysis for body fat mass and eating behavior, accompanied by inversely health−related *Parabacteroides* genus, remained largely significant [42,43], whereas mechanistic insights into how specific *Parabacteroides* modulate the eating behavior of human and weight status are still limited. *Lactobacillus* and *Candidatus Saccharimonas* genera were both or separately negatively related to inguinal adipose tissue weight or body weight in HFD−induced mice. This is in line with *Lactobacillus* significantly reducing the body weight gain, adipose index and serum TG level, and through the regulation of adipocytokines that displayed strong anti−obesity effects on high−fat−diet−fed mice [44]. Combining our results in Figure 6B, these findings fully verified that NOB improved certain gut microbiota such as *Lactobacillus* to inhibit fat accumulation. Recent reports have shown that policosanol-promoted lipolysis and thermogenesis processes are correlated with the decreasing level of *Lactobacillus* and *Candidatus Saccharimonas* [45]. Overall, these results suggest that *Lactobacillus* may be promising functional materials in healthy diets. *Enterococcus* and *Helicobacter* genus were extremely positively correlated with inguinal adipose tissue weight in HFD−NOB mice (*p* < 0.01), yet displayed an inverse correction both in chow and HFD mice. The result is in accordance with the understanding that obese individuals had a higher risk of *Helicobacter* infection than lean individuals [42]. Based on the above results, we observed that the underlined NOB plays a positive role in obese individuals by increasing the beneficial gut microbiota abundance.

## 5. Conclusions

In summary, our study investigated the effect and mechanism of nobiletin on fat deposition in high−fat−diet−fed mice. Our data showed that intragastrical administration of NOB attenuates HFD-induced obesity, hyperlipidemia and lipid metabolic disorder. Next, a 16S amplicon sequencing analysis of NOB−treated mice feces improved the gut microbiota composition, function and metabolism pathway and NOB alleviated obesity partially via the modulation of gut microbiota. Therefore, NOB holds promise for obesity and its related metabolic disorders. Further investigation of the specific bacterial species will provide more valuable information for the exact underlying mechanisms and merits of NOB.

## Figures and Tables

**Figure 1 genes-14-01062-f001:**
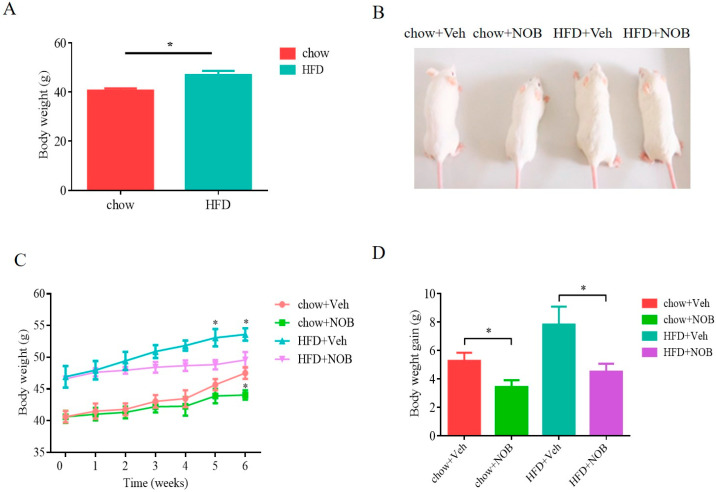
NOB attenuates HFD−induced obesity. (**A**) Bodyweights of mice fed with chow or HFD (**B**) Images of Kunming mice treated with vehicle or NOB for 6 weeks. (**C**) Body weight of chow + Veh, chow + NOB, HFD + Veh and HFD + NOB. (**D**) Body weight gain. Graph bars in (**A**,**C**,**D**) marked with * stand for significant results; one−way ANOVA and Student’s t−test for analyzing statistical differences (** p <* 0.05), n ≥ 6 per group.

**Figure 2 genes-14-01062-f002:**
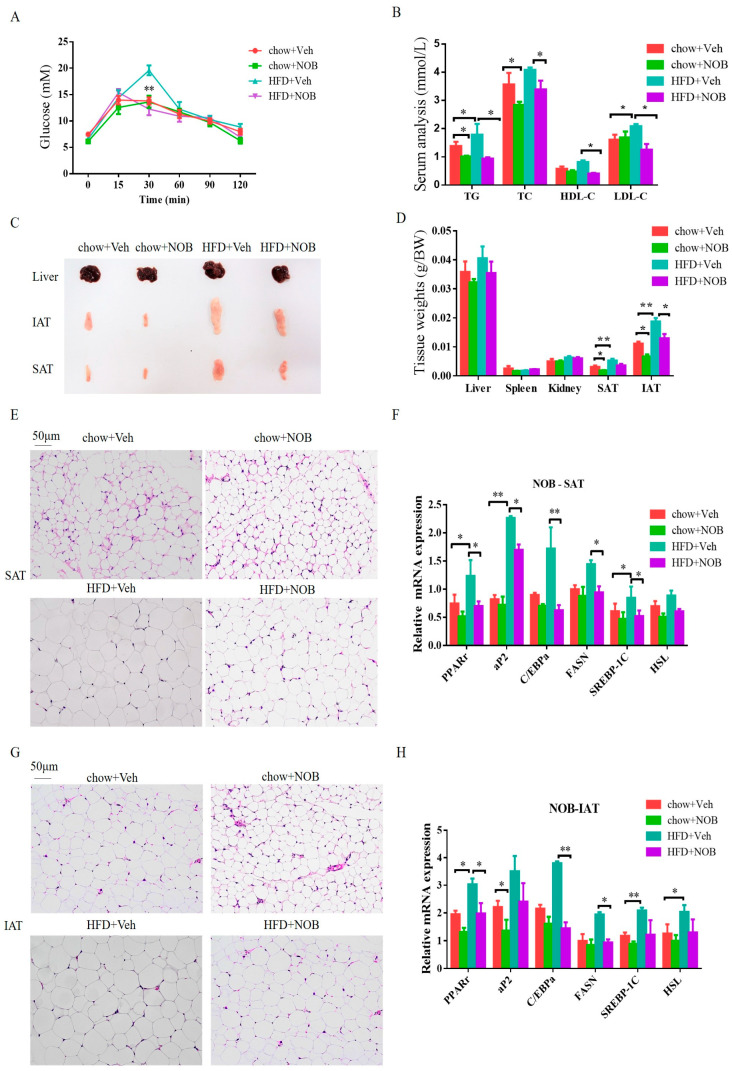
NOB improves glucose tolerance and lipid metabolic disorder in HFD−fed mice. (**A**) GTT (Glucose tolerance test) results. (**B**) The contents of TC (total cholesterol), TG (total glyceride), HDL−C (high−density lipoprotein cholesterol) and LDL−C (low-density lipoprotein cholesterol). (**C**) The patterns of the liver, IAT and SAT. (**D**) The tissue weight of chow and NOB−treated mice. (**E**) The sections of H&E staining from SAT. Scale bar, 50 μm. (**F**) The mRNA level of *PPARγ*, *ap2*, *C/EBPα*, *FASN*, *SREBP*−*1C* and *HSL* in SAT. (**G**) The sections of H&E from IAT. Scale bar, 50 μm. (**H**) The mRNA level of *PPARγ*, *ap2*, *C/EBPα*, *FASN*, *SREBP*−*1C* and *HSL* in IAT. Data are shown as mean ± SEM. Significance test between groups was conducted by utilizing Student’s t−test or ANOVA (* *p* < 0.05, ** *p* < 0.01).

**Figure 3 genes-14-01062-f003:**
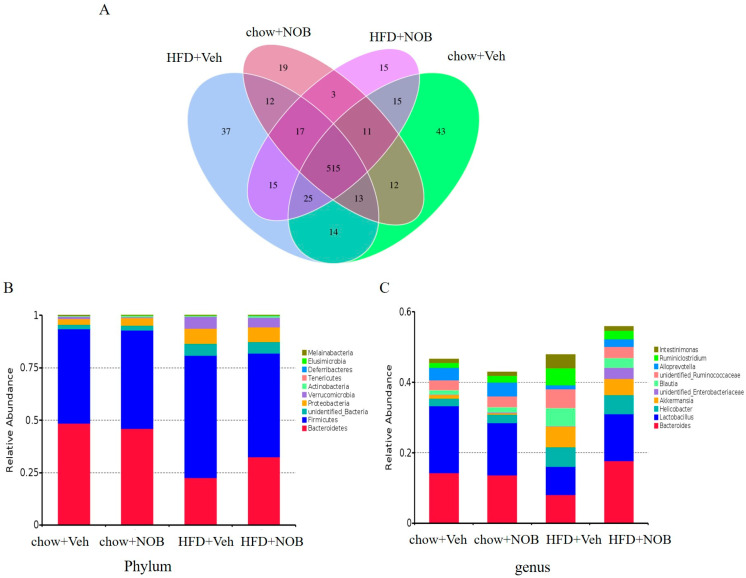
NOB modifies the composition of intestinal flora in HFD−induced mice. (**A**) The Venn diagram for the distribution of treatment by vehicle or NOB. (**B**) The proportion of gut microflora at phylum level per group. (**C**) The proportion of gut microflora at genus level per group.

**Figure 4 genes-14-01062-f004:**
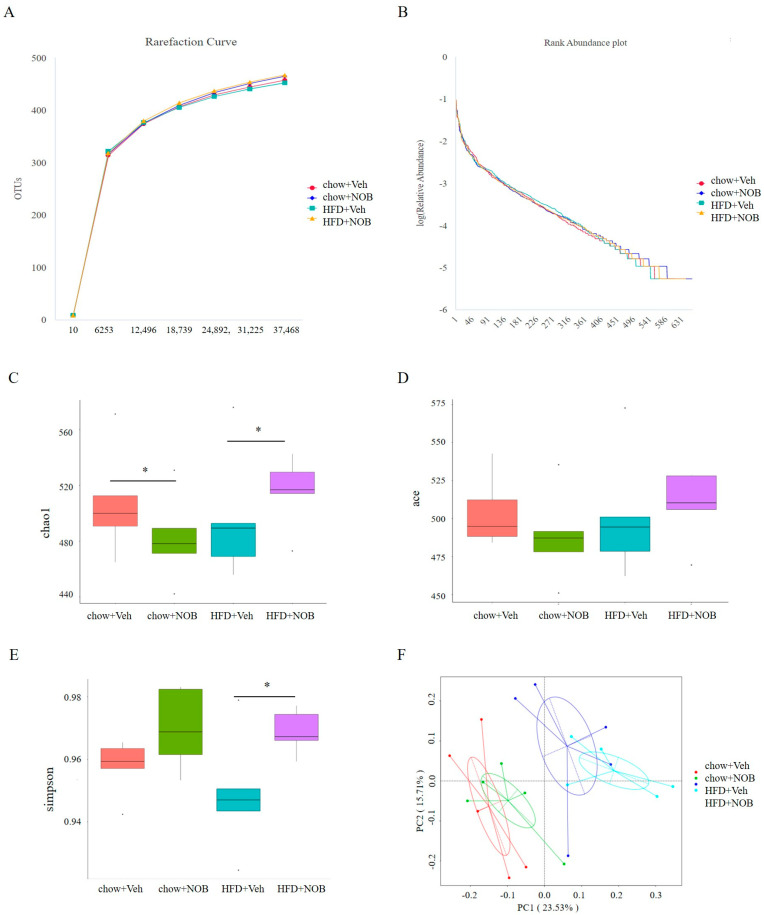
NOB improves intestinal flora diversity in HFD-induced mice. (**A**) The curve of rarefaction. (**B**) The curve of rank abundance. (**C**) Chao1 index in the ɑ−diversity analysis. (**D**) ACE index in the ɑ−diversity analysis. (**E**) Simpson index in the ɑ−diversity analysis. (**F**) The analysis Bray−Curtis−based PCoA plot from each sample. Points are from individual mice. Significant test results based on one−way ANOVA along with Student’s t-test (* *p* < 0.05).

**Figure 5 genes-14-01062-f005:**
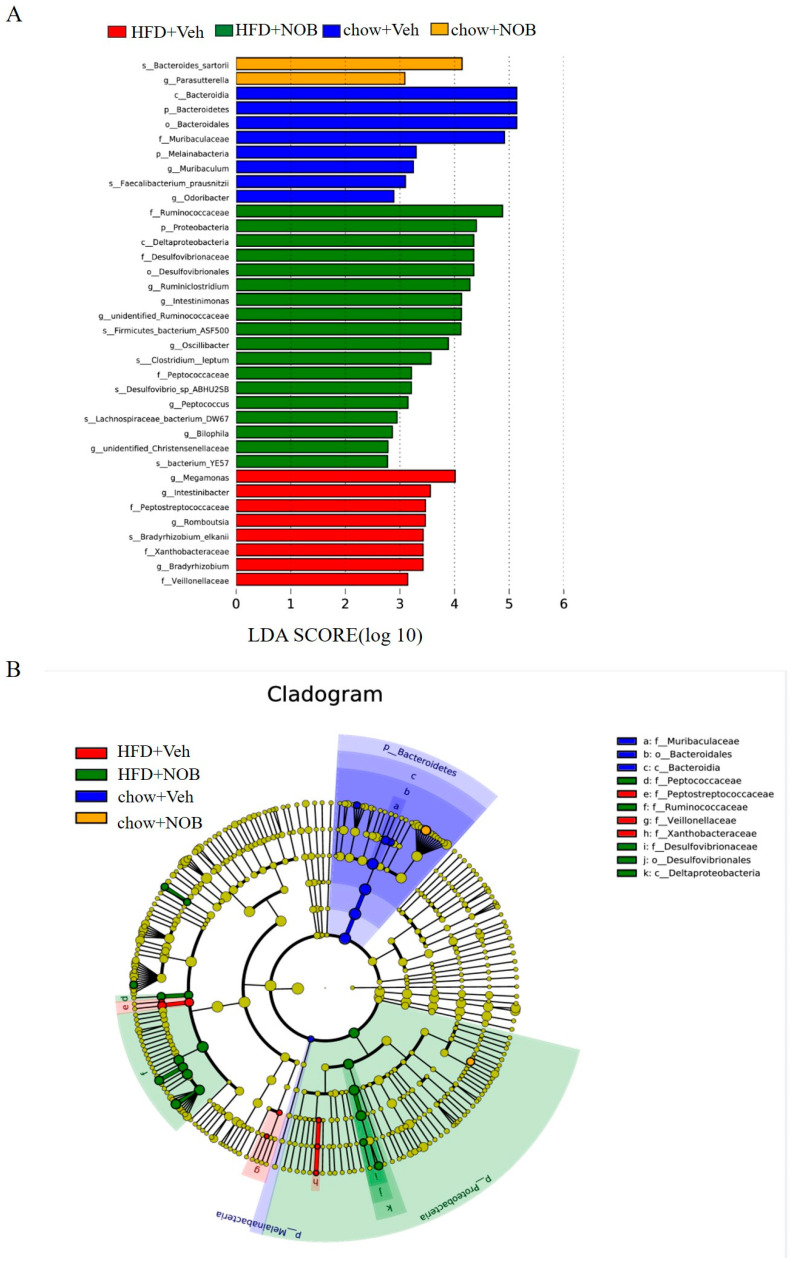
NOB modifies gut microbial and metabolism pathways. (**A**) The different biomarker taxons by LEfSe analysis (LDA > 2). (**B**) Cladogram obtained from LEfSe analysis and the levels from inner to outer rings stand for phylum, class, order, family and genus.

**Figure 6 genes-14-01062-f006:**
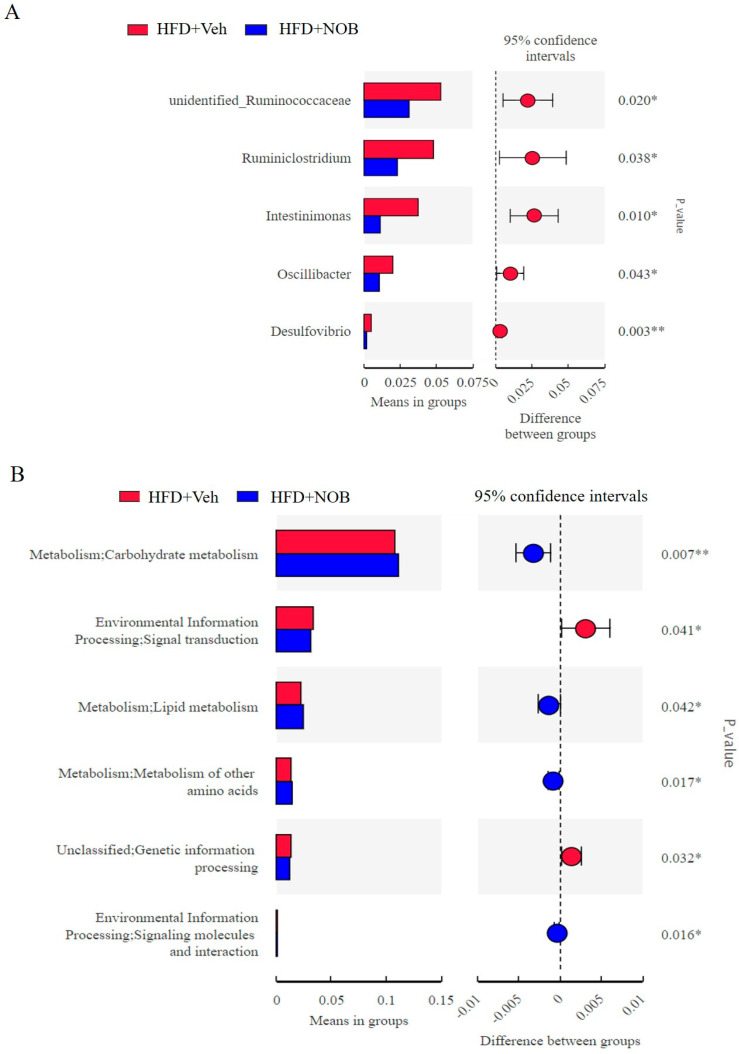
NOB modifies metabolism pathways. (**A**) Taxon constitution with the top 5 typical bacteria determined at genus level by Wilcoxon rank −sum test (* *p* < 0.05, ** *p* < 0.01). (**B**) Metabolic pathways were predicted by PICRUSt analysis with Student’s t−test (* *p* < 0.05, ** *p* < 0.01).

**Figure 7 genes-14-01062-f007:**
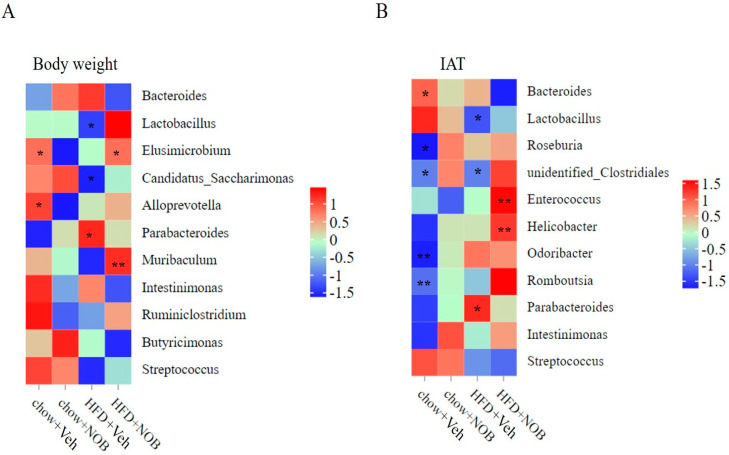
The correlation is between adipose tissue and intestinal flora in chow or HFD−fed mice. The different relevance between the intestinal flora and body weight (**A**) as well as adipose tissue (**B**) in chow or HFD-fed mice under NOB treatment (** p* < 0.05*, ** p* < 0.01).

**Table 1 genes-14-01062-t001:** Gene Sequences for qPCR.

Gene	Forward primers(5′-3′)	Reverse primers(5′-3′)
GAPDH	TGGCCTTCCGTGTTCCTAC	GAGTTGCTGTTGAAGTCGCA
C/EBPα	TGGACAAGAACAGCAACGAG	TCACTGGTCAACTCCAGCAC
ap2	AAGAAGTGGGAGTGGGCTTTG	CTCTTCACCTTCCTGTCGTCTG
PPAR-γ	CCAAGAATACCAAAGTGCGATCA	CCCACAGACTCGGCACTCAAT
SREBP-1c	GGAGCCATGGATTGCACATT	GGCCCGGGAAGTCACTGT
FASN	GCTGCGGAAACTTCAGGAAAT	AGAGACGTGTCACTCCTGGACTT
ATGL	TTCGCAATCTCTACCGCCTC	AAAGGGTTGGGTTGGTTCAG
HSL	GCTGGGCTGTCAAGCACTGT	GTAACTGGGTAGGCTGCCAT
LPL	CCAATGGAGGCACTTTCCA	TGGTCCACGTCTCCGAGTC

## Data Availability

The corresponding author or first author will provide data supporting this research study upon reasonable request.

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
