# Peer review of "Modulation of Fat Deposition–Gut Interactions in Obese Mice by Administrating with Nobiletin"

_genes, 2023, doi:10.3390/genes14051062_

Round 1

Reviewer 1 Report

Review Report

Title : Modulation of fat deposition-gut interactions in obese mice by  administrating with nobiletin (genes-2368843)

Journal: Genes

Authors: Cunzhen Zhao, Jiahua Guo, Chunyu Du, Yongjie Xu *

Introduction

Line 44 : Replace “attached” with “attracted”.

Line 70 : Replace “under” with “in”.

Line 77: Replace “male Kunming weighing 18-20g mice” as “Male Kunming mice weighing”

Line 77: Why 24-32 numbers? Since the experiment is completed, you can mention precise number of animals.

Line 99: Replace “feces samples” with “fecal samples”.

Line111: Delete small. Replace “Glucose meter” with “Glucometer”

Line 149: Change the title as “Statistical analysis”

Result

Fig 2 : The histopathology sections (H&E) are not clear

Fig 6 B: It is shown that fat metabolising enzymes are increased with NOB than HFD. What can be the possible reason for this?

Discussion

Line 430 “Candidatus_Saccharimonas” please correct

Reviewer 2 Report

In the manuscript genes-2368843, “Modulation of fat deposition-gut interactions in obese mice by administrating with nobiletin”, Zhao and colleagues try to investigate possible therapeutic role of NOB in a HFD model of obesity. In the last decade, major contributions have been made in this field but the lack of treatment make the study of therapeutics targets very important in this field. Moreover, the importance of the microbiota to the development of several diseases makes this topic, a highlight to researchers. With a mix of manual (mice-diet) and computational (analysis) work they conclude the potential use of NOB to prevent obesity. Although, the structure and narrative of the article is very well organized, several points may be considered:

Major points

-       HFD models are very well studied and established. Several reviews show different timing and diets (brands) to produce high quality results in this model. However, the authors, only use 6 weeks of feeding (maybe because we don’t know how many time the mice where on a diet before NOB administration). For that reason:

o   1. Authors should explain very clearly the timing for diet

o   2. They used Kunming mice. To prove their results, they should do the experiment with another strain: C57BL6j (for example).

-       As the reader don’t know how much time mice are on diet before NOB administration, it seems that the bodyweight in chow animals is extremely high when you compare this study with the bibliography. That problem should be addressed.

-       For this part of the study, several parameters are missing when you study fat in a HFD model. Crown like structures, data for inflammation, cell death and immune infiltration should be also required.

-       Figure 4 and 5: colors did not much the colors first use in fig1 and 2 for the different groups. These colors should be consistent for the whole article.

-       The second part of the study is very descriptive about the different microbiota associated a HFD and NOB administration. However, the meaning of these results are only descriptive, without to study further if:

o   Probiotics, or FMT transplantation could also alleviate mice from obesity induced in a HFD model

o   Or, the supplementation with metabolites associated to the pathways with higher changes, could be beneficial for the progression of the obesity.

Any of these options should be considered to study, to correlate data from microbiota to benefit or not for the progression of HFD-induced obesity

Minor points

-       The age of the mice is missing in methods.

-       The authors said that they divide the mice in 2 groups (+/- HFD). When the bodyweight is significantly different, they sub divide the groups in vehicle or NOB administration. To better knowledge of the methodology, they should add a time between the beginning of the diet until the time they start with NOB administration. Moreover, they name the vehicle group NaCl instead CMC-Na (the solution used for NOB administration-vehicle). Maybe, changing NaCl for Veh (vehicle) could be better.

-       In 2.4: authors do not name what is GTT, TC and TG. Also, they said: blood lipids, cholesterol and other indicators……… Every one of the indicators should be added to this paragraph (like HDL-C or LDL-C) with the reference of the kits used for their detection.

-       Fig legend 1: A and B they did not introduce properly the graphic showed. A) The body weight of mice were treated with chow or high-fat diet. Instead, the tittle should be like this: Bodyweights of mice feed a chow or HFD. The figure did not show any **p<0,01. Should be removed.

-       Fig 2: in E and G, a line with the type or fat should be added above the pictures.

-       Fig legend 2: what is the meaning of the arrow in the pictures? Arrow was not indicated at the text neither the figure legend.

-       L210-212 should be re-wrote

-       Figure legend 3: level per group

-       L232-235: did not match the significance at the picture.

-       Fig legend 4: Shannon index, should be changed for Simpson, or give an explanation of Shannon index in the text.

-       L261: LEfSe - should be explained or if is a program, should go with ( …)

-        

Very poor quality of the language. It should be revised for an expert, to help to improve the article.

for example in : 3.4 NOB administration ......

Authors said several times the "index of Chao1" , the "index of Simpson", other times Chao1 index. Be consistent!

There so many present phrases instead the past sentences. (is instead was-L189). Use of -ing instead -ed (indicating -L185)

L202: induces the changes (the should be eliminated)

Figure legend 3: at phylum in indicated group should be re-write as : at phylum level per group.

Round 2

Reviewer 2 Report

Although the improve the manuscript, there still some issues:

- Fig1A: It was my mistake type feed instead fed. It is the past form: "fed"

- Fig1B: this phrase doesn't represent the picture. Modify it.

L221-223: this phrase did not change from the original manuscript, and it is very incoherent. Needs more information.

Minor review 

Author Response

- Fig1A: It was my mistake type feed instead fed. It is the past form: "fed"

Response:Thanks! We have amend the Fig1A legend with “Bodyweights of mice fed with chow or HFD”

- Fig1B: this phrase doesn't represent the picture. Modify it.

Response:Thanks! We have rephrased the Fig1B legend with “Images of Kunming mice treated with vehicle or NOB for 6 weeks.

L221-223: this phrase did not change from the original manuscript, and it is very incoherent. Needs more information.

Response: Thanks for your suggestion. This was due to our writing error and we amended the sentence as follows: The ratio of Firmictutes to Bacteroidetes was higher in obese individuals and increased abundance of Bacteroidetes related to the percentage loss of weight.